# Single-transistor organic electrochemical neurons

Junpeng Ji [1], Dace Gao [1], Han-Yan Wu[1], Miao Xiong[1], Nevena Stajkovic[2,3], Claudia Latte Bovio[4,5], Chi-Yuan Yang [1], Francesca Santoro [2,3,4], Deyu Tu [1] & Simone Fabiano [1] ✉

Neuromorphic devices that mimic the energy-efficient sensing and processing capabilities of biological neurons hold significant promise for developing bioelectronic systems capable of precise sensing and adaptive stimulus-response. However, current silicon-based technologies lack biocompatibility and rely on operational principles that differ from those of biological neurons. Organic electrochemical neurons (OECNs) address these shortcomings but typically require multiple components, limiting their integration density and scalability. Here, we report a single-transistor OECN (1T−OECN) that leverages the hysteretic switching of organic electrochemical memtransistors (OECmTs) based on poly(benzimidazobenzophenanthroline). By tuning the electrolyte and driving voltage, the OECmTs switch between high- and low-resistance states, enabling action potential generation, dynamic spiking, and logic operations within a single device with dimensions comparable to biological neurons. The compact 1T−OECN design (~180 $\mu m^2$ footprint) supports high−density integration, achieving over 62,500 neurons/$cm^2$ on flexible substrates. This advancement highlights the potential for scalable, bio-inspired neuromorphic computing and seamless integration with biological systems.

Neurons are essential for sensory perception and information processing, enabling organisms to adapt to dynamic environmental stimuli. They communicate through ion-driven processes and action potentials (spikes), forming the basis of complex neural computations[1]. Inspired by this natural efficiency, neuromorphic devices seek to replicate neuronal functionality, particularly event-driven spiking mechanisms, to enable real-time, energy-efficient processing[2]. These devices have broad potential for applications ranging from bioelectronics[3] and robotics[4] to brain-computer interfaces[5], offering a pathway toward seamless integration with biological systems.

While traditional artificial neuron technologies, including those based on silicon and other inorganic materials[6–9], have successfully advanced the field of neuromorphic computing, they face inherent challenges in emulating the dynamic and ion-regulated processes of biological neurons. These technologies often rely on rigid materials and purely electronic mechanisms, which can limit biocompatibility and hinder their direct interaction with biological systems. Despite their many achievements, addressing these challenges is essential for developing neuromorphic devices that closely mimic natural neural behaviors.

Organic electrochemical neurons (OECNs)[10–14], built from organic electrochemical transistors (OECTs), offer a transformative approach to bridging biology and electronics. OECTs use ion-driven doping in organic mixed conductors[15–18] to enable dynamic signal processing

[1]Laboratory of Organic Electronics, Department of Science and Technology, Linköping University, Norrköping, Sweden. [2]Institute of Biological Information Processing IBI-3 Bioelectronics, Forschungszentrum Jülich, Jülich, Germany. [3]Neuroelectronic Interfaces, Faculty of Electrical Engineering and IT, RWTH Aachen, Aachen, Germany. [4]Tissue Electronics, Center for Advanced Biomaterials for Healthcare, Istituto Italiano di Tecnologia, Naples, Italy. [5]Dipartimento di Ingegneria Chimica, dei Materiali e della Produzione Industriale, Università degli Studi di Napoli Federico II, Naples, Italy. ✉e-mail: simone.fabiano@liu.se

that closely mimics biological systems. Their high conformability, biocompatibility, and ability to operate at low voltages make OECTs ideal for bio-interfacing applications[11,19]. These properties have propelled OECTs into applications such as artificial synapses[20–22], spiking neurons[23–26], and neural signal transmission systems[10]. OECNs, specifically, stand out for their ability to mimic various neuronal functions, including signal integration, threshold-dependent spiking, and ion modulation—capabilities essential for the development of event-based sensors. Despite their promise, current OECN designs typically require multiple components, including OECTs, resistors, capacitors, and amplifiers, to replicate neuronal behaviors. This reliance on complex circuits limits their scalability, integration density, and operational simplicity. Simplifying these designs while retaining functionality is critical for advancing OECNs toward practical applications, such as dense neuromorphic networks and bioelectronic systems.

Here, we introduce single-transistor organic electrochemical neurons (1T−OECNs) based on poly(benzimidazobenzophenanthroline) (BBL)−based organic electrochemical memtransistors (OECmTs). These BBL−based OECmTs leverage ion-tunable antiambipolarity and asymmetric transient response to achieve voltage-dependent, hysteretic transitions between high- and low-resistance states. This behavior enables essential neuromorphic functions—including action potential generation, dynamic spiking, and neuromorphic logic—within a single device with dimensions comparable to biological

neurons (<50 μm). 1T−OECNs emulate key neuronal behaviors such as voltage-dependent thresholding, spike generation, and frequency adaptation without requiring additional passive components. They exhibit 17 distinct neural features and respond to both chemical and electrical stimuli. Integrated with resistive pressure sensors, they function as artificial afferent nerves, transducing mechanical stimuli into neuromorphic pulse trains. Beyond neuronal dynamics, BBL−OECmTs also operate as organic electrochemical synapses, supporting excitatory and inhibitory responses along with spike-timing-dependent plasticity. Their compact design, with a footprint as small as 177 μm², enables high−density integration with neuron densities exceeding 62,500 neurons/cm². These versatile devices support universal logic operations, advancing in-memory computing and enabling real-time, low-power neuromorphic processing for applications in bioelectronics and next-generation neuromorphic networks.

## Results

### Antiambipolarity and switching dynamics of a BBL−OECmT

The antiambipolar transfer characteristics of BBL−based OECTs with a 0.1 M NaCl electrolyte are shown in Fig. 1a. As the gate voltage ($V_G$) increases, the BBL channel transitions reversibly through three states: undoped (low-conductance), doped (high-conductance), and highly doped (low-conductance), resulting in a finite window of electrical conductivity[27]. This window can be tuned by modifying the electrolyte

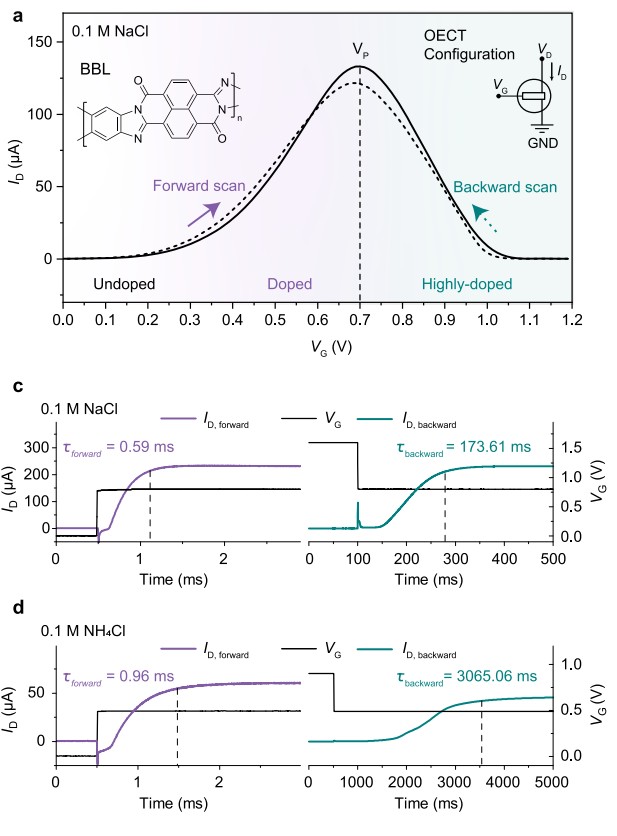

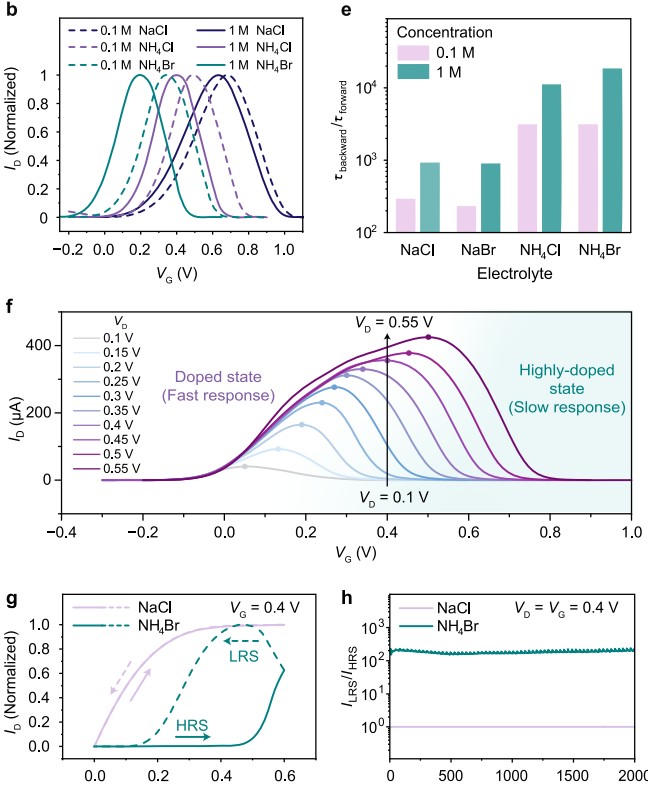

**Fig. 1 | Electrical characteristics of BBL−OECmTs. a** Antiambipolar transfer characteristics of a BBL−based OECT. The peak voltage ($V_P$) corresponds to the gate voltage ($V_G$) value at which the drain current ($I_D$) reaches a maximum. As $V_G$ increases toward $V_P$, BBL transitions from an undoped state to a doped state (forward scan). At $V_G > V_P$, BBL enters a highly doped state. Inset: OECT diagram and BBL chemical structure. **b** Influence of electrolyte concentration and ion type on the antiambipolar characteristics of BBL OECTs. **c, d** Transient responses of BBL OECTs using 0.1 M NaCl (**c**) and 0.1 M NH₄Cl (**d**) as electrolytes. Black curves represent the gate voltage changes during forward (increasing $V_G$) and backward (decreasing $V_G$) scans. Purple and cyan curves show corresponding current responses. The 90% turn-on times are marked as $\tau_{forward}$ and $\tau_{backward}$ for forward

and backward scans, respectively. **e** Comparison of turn-on response times for BBL OECTs with different electrolytes and concentrations. **f** Transfer curves of BBL OECTs measured under different drain voltages ($V_D$), using 1 M NH₄Br as the electrolyte. Points on each curve mark $V_P$. Left of $V_P$: doped/fast response regime; right of $V_P$: highly doped/slow response regime. **g** Output curves at a fixed sweep rate of 0.2 V s⁻¹ using 1 M NH₄Br and 1 M NaCl as electrolytes. $V_D$ is swept from 0 to 0.6 V and back. Only with NH₄Br does the device show memtransistor behavior: transitioning from a high-resistance state (HRS) to a low-resistance state (LRS) upon returning to 0 V. **h** Endurance of the $I_{LRS}/I_{HRS}$ ratio over 2000 consecutive cycles at $V_D = V_G = 0.4$ V for devices operated in 1 M NH₄Br and 1 M NaCl.

composition and driving voltage. Replacing NaCl with an electrolyte containing protic cations, such as $NH_4Cl$, reduces the peak-current voltage ($V_p$) from 0.68 V to 0.49 V (Fig. 1b), consistent with previous reports[23]. The operational voltage range of these devices can be further lowered by using $Br^-$ anions. Replacing $NH_4Cl$ with $NH_4Br$ modifies the electrochemical behavior of the Ag/AgCl paste gate electrode by altering potential dynamics at the electrode/electrolyte interface[28]. This results in a reduced effective gate potential and shifts the OECT transfer curves toward lower $V_G$ (Supplementary Fig. 1). Consequently, $V_p$ shifts from 0.49 V for $NH_4Cl$ to 0.34 V for $NH_4Br$ (Fig. 1b). Increasing the electrolyte concentration from 0.1 M to 1 M further shifts the current peak towards lower voltage biases, with $V_p$ reaching values as low as 0.20 V for 1 M $NH_4Br$ (Fig. 1b).

Thus, BBL-based OECTs can be switched on by either increasing $V_G$ toward $V_p$ (forward scan) or decreasing $V_G$ toward $V_p$ (backward scan). However, the transient responses of these two scans ($\tau_{forward}$ and $\tau_{backward}$, defined as the time required for the current to reach 90% of its final value) differ significantly. Specifically, $\tau_{backward}$ can be >300× longer than $\tau_{forward}$ when 0.1 M NaCl is used as the electrolyte (Fig. 1c). This difference increases to over 3000× with 0.1 M $NH_4Cl$ (Fig. 1d) and exceeds 10,000× with 1 M $NH_4Cl$ or $NH_4Br$ electrolytes (see Fig. 1e and Supplementary Figs. 2 and 3). We attribute the slower response observed during the backward scan to the formation of multiply charged species with reduced mobility in the highly doped state[27], which increases the effective RC time constant and leads to longer current transients. This effect is further enhanced by ions like $NH_4^+$, which strongly interact with the BBL backbone (e.g., through hydrogen bonding[23]), thereby hindering ionic mobility and extending the transient response time.

Additionally, increasing the drain voltage ($V_D$) raises the peak current and shifts $V_P$ to higher values due to the different doping levels at the source and drain electrodes (Fig. 1f). This leads to a change in the doping state even when $V_G$ is fixed. As indicated by the black arrow in Fig. 1f, when $V_G$ is held at 0.4 V, scanning $V_D$ from low to high values transitions the channel from a highly doped state (right side of the Gaussian current curve relative to $V_p$) to a doped state (left side of the Gaussian current curve relative to $V_p$) as $V_D$ increases from 0.1 V to 0.55 V.

This asymmetric switching behavior, combined with the tunable antiambipolar characteristics of BBL, enables the development of OECmTs, where the channel transitions hysteretically between a high-resistance state (HRS, undoped or highly doped) and a low-resistance state (LRS, doped) depending on $V_G$, $V_D$, and the scan rate. Supplementary Note 1 provides a detailed explanation of the mechanism underlying the hysteretic behavior of BBL-OECmTs. Figure 1g shows the output characteristics of a BBL-based OECmT with 1 M $NH_4Br$ electrolyte, recorded at $V_G = 0.4$ V while sweeping $V_D$ between 0 and 0.6 V at a scan rate of 0.2 V s$^{-1}$. The device exhibits pronounced current hysteresis, consistently switching between LRS and HRS without significant current decay (see Supplementary Note 2 for SPICE simulations). As $V_D$ increases from 0 to 0.45 V, $I_D$ rises gradually due to the slow response of the device in the highly doped state. Beyond $V_D = 0.45$ V, $I_D$ increases more rapidly as the device transitions into the doped state, where it responds faster to change in $V_D$. When $V_D$ is swept back to 0 V, $I_D$ initially remains high and then decreases gradually as the device re-enters the highly doped state. The magnitude of the hysteresis changes with different scan rates (Supplementary Fig. 5), a characteristic feature of memristive elements[29,30]. Notably, when $V_D = V_G = 0.4$ V, the ratio of HRS current ($I_{HRS}$) to LRS current ($I_{LRS}$) reaches its maximum value (-150) and remains stable over 2000 cycles (Fig. 1h). In contrast, BBL-based OECmTs with NaCl electrolyte show no hysteretic behavior at this $V_G$ and scan rates <0.2 V s$^{-1}$ (Fig. 1g) and require $V_G > 0.7$ V and scan rates > 0.2 V s$^{-1}$ to exhibit only a slightly appreciable hysteresis (see Supplementary Figs. 4b and 5b). For this reason, we focus exclusively on OECmTs comprising $NH_4Br$ as the electrolyte for the rest of this study. Overall, these BBL-based OECmTs operate at much lower voltages than their inorganic counterparts (e.g., those based on 2D materials[31]) and leverage the advantage of three-terminal architectures, making them ideal for biomimetic neuromorphic applications.

## Action potential generation in a 1T-OECN

In biological neurons, action potentials are generated through the coordinated timing of rapid and delayed ion channel activation (Fig. 2a). Upon sufficient stimulation, voltage-gated sodium ($Na^+$) channels rapidly activate, allowing $Na^+$ influx and causing membrane depolarization. At peak membrane potential, the $Na^+$ channels inactivate. This is followed by a delayed activation of potassium ($K^+$) channels, which enable $K^+$ efflux and restore the membrane potential through repolarization. This sequential activation ensures efficient signal propagation and resets the membrane potential for subsequent signaling. Similarly, BBL-based OECmTs integrate two key neuronal phenomena—voltage-dependent negative differential resistance and delayed channel activation—within the same device, allowing it to emulate neuronal functions using a single-OECT layout. As shown in Fig. 2b, when an external stimulation current ($I_{IN}$) is applied to the drain of the OECmT, while a constant $V_G$ is applied at the gate, a membrane voltage ($V_{MEM}$) equivalent to $V_D$ is measured at the drain, with the source grounded. In this system, the initial charging of the intrinsic OECmT capacitance causes an increase in $V_{MEM}$, corresponding to the depolarization process (Fig. 2c, left). At this stage, when $V_D < 0.45$ V and $V_G = 0.4$ V, the channel is highly doped, exhibiting high resistance, which prevents current leakage.

The increase in $V_{MEM}$ causes the conductance state of the OECmT channel to change. As shown in Fig. 2d, when $V_D$ exceeds 0.45 V and at $V_G = 0.4$ V, the channel transitions from a highly doped state/HRS to a doped state/LRS, corresponding to the condition $V_P - V_G > 0$ (i.e., $V_P > V_G$). At this stage, the channel current increases with a delay, allowing current to leak through the channel ($I_{LEAK}$) and causing $V_{MEM}$ ($V_D$) to decrease. This behavior corresponds to the repolarization phase, mimicking the delayed opening of $K^+$ channels (Fig. 2c, middle). When $V_D$ decreases below 0.45 V, the OECmT channel re-enters the highly doped state and begins to slowly close. As $V_{MEM}$ decreases to the resting potential and $I_{LEAK}$ remains greater than $I_{IN}$, $V_{MEM}$ continues to drop below the resting potential, corresponding to the hyperpolarization process (Fig. 2c, right). Subsequently, the channel fully transitions into the highly doped state/HRS, where $I_{LEAK} < I_{IN}$, and the input current is no longer sufficient to effectively pass through the channel. At this stage, the intrinsic OECmT capacitance begins to recharge, causing $V_{MEM}$ to increase again (see Supplementary Note 3 for SPICE simulations). This cycle repeats as long as the external current continues to be applied. As shown in Fig. 2e, during each spike, the OECmT demonstrates current behavior analogous to biological neurons, including a rapid charging current ($I_{CHARG}$) activation and a delayed leakage current ($I_{LEAK}$) activation.

## Neural features and logic capabilities of the 1T-OECNs

Hence, the operation of the 1T-OECNs is defined by three key variables: the internal capacitor charge, the doping state transition speed, and the gate-controlled doping state. Together, these factors create a complex system that governs the device's behavior and enables tunable neural features resembling those found in biological neurons. By adjusting $I_{IN}$ and $V_G$, we successfully replicated 17 distinct neural features using a single OECT (see Supplementary Note 4 and Supplementary Table 1 for a detailed explanation of the various spiking modes). Compared to other spiking neuron technologies made from different material classes, the 1T-OECN combines high biological plausibility with low circuit complexity (Fig. 3a and Supplementary Table 2), closely mimicking the functionality of biological neurons.

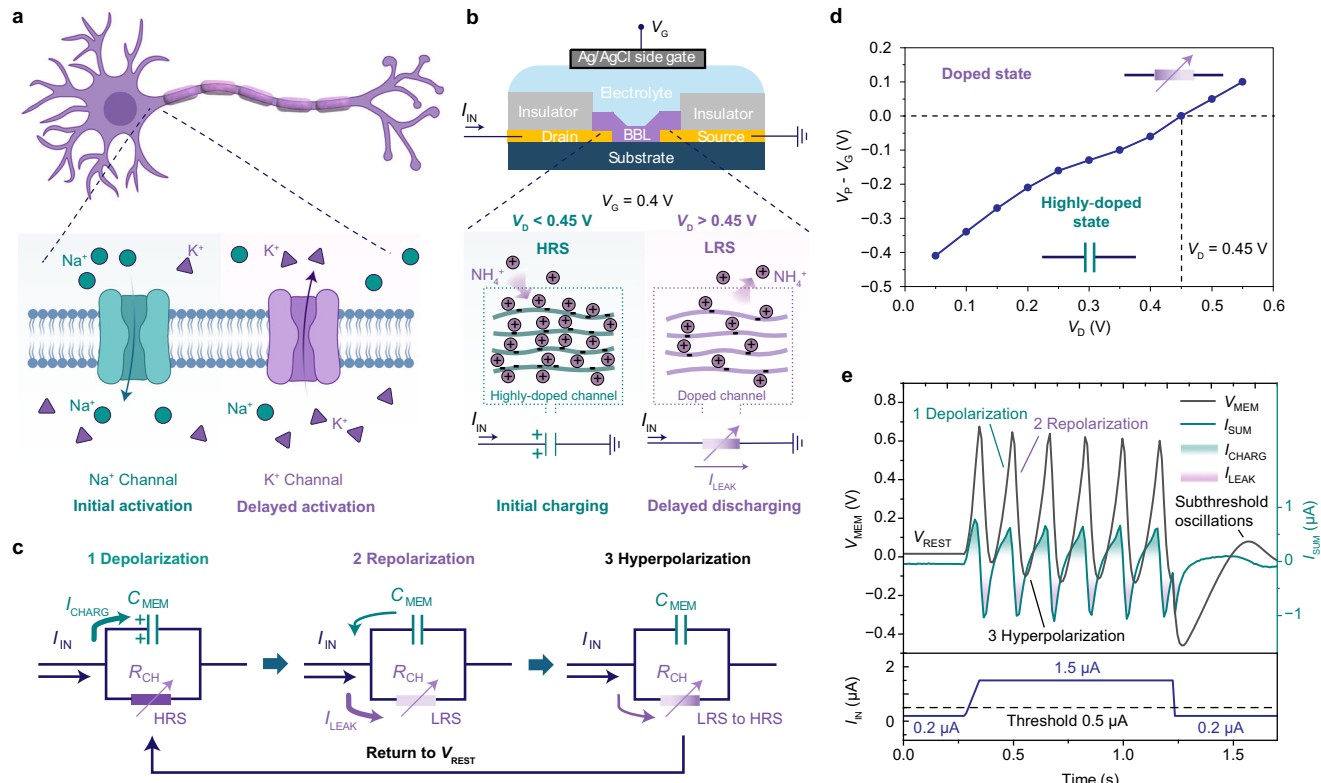

**Fig. 2 | Action potential generation mechanism in a 1T–OECN. a** Schematic of a biological neuron showing voltage-gated Na⁺ and K⁺ ion flows across the membrane that drive action potential generation. **b** Structure of the BBL-based 1T–OECN, illustrating its analogy to biological neurons: initial capacitive charging mimics rapid Na⁺ channel activation, while delayed discharging corresponds to delayed activation of K⁺ channels. **c** Circuit diagram of the BBL-based 1T–OECN, mapping the depolarization, repolarization, and hyperpolarization phases to those of biological neurons. **d** Difference between $V_G$ and $V_P$ as a function of $V_D$ for a

BBL–OECmT using 1 M NH₄Br. $V_G$ is fixed at 0.4 V during the measurements. When $V_D > 0.45$ V, the channel transitions from a highly doped to a less doped state, corresponding to the condition $V_P–V_G > 0$. **e** Representative action potentials in a BBL-based 1T–OECN, showing fast activation of the charging current ($I_{CHARG}$) followed by delayed onset of the leakage current ($I_{LEAK}$), mimicking Na⁺/K⁺ channel dynamics. $I_{SUM}$ represents the total current at $V_{MEM}$, calculated as the applied stimulation current ($I_{IN}$) minus $I_{LEAK}$.

When a constant input current exceeding the threshold is applied, the 1T–OECN exhibits tonic spiking (Supplementary Fig. 11a). For channel length $L = 10\,\mu m$ and width $W = 20\,\mu m$, an $I_{IN} = 40$ nA induces tonic spiking with energy consumption as low as 4.7 nJ/spike (Supplementary Fig. 11b, c). The energy consumption per spike is projected to reach pJ level by further reducing the W/L ratio (see Supplementary Fig. 11d). The 1T–OECN demonstrates all three classes of excitability (Fig. 3b) and supports features like refractory period plasticity (Supplementary Fig. 12a), phasic spiking (Supplementary Fig. 12b), mixed-mode spiking (Supplementary Fig. 12c), spike frequency adaptation (Supplementary Fig. 12d), random firing (Supplementary Fig. 13a), damped spiking (Supplementary Fig. 13b), and beyond-threshold damped oscillations (Supplementary Fig. 13c). Stochastic spiking is also achieved in these 1T–OECNs, where excessive noise induces fluctuations around the base frequency (see Fig. 3c). Additional neural features, such as integration, refractoriness, resonance, and accommodation (Supplementary Fig. 14), match the capabilities of complex Hodgkin–Huxley circuits but are achieved here using a single transistor.

Compared to other artificial spiking neuron technologies, 1T–OECNs uniquely support dual-mode responsiveness to chemical and electrical signals to modulate spiking behavior. Different ionic species and concentrations influence the device's working state, as shown in Fig. 3d. For example, with $I_{IN} = 3\,\mu A$, the 1T–OECN operates in mixed-mode with 0.1 M NH₄Br but transitions to tonic spiking with 0.1 M NH₄Cl. Increasing $I_{IN}$ to 5 μA induces tonic bursting with larger voltage amplitudes. Beyond ion regulation, spiking frequency and

dynamics can be precisely controlled by varying $I_{IN}$, $V_G$, and source voltage ($V_S$). For instance, at $V_G = 0.45$ V, increasing $I_{IN}$ from 1.8 μA to 30 μA shifts spiking from mixed-mode (-1.4 Hz) to tonic spiking (32 Hz) (Supplementary Fig. 15). Adjusting $V_G$ and $V_S$ offers further control over the spiking frequency (Supplementary Fig. 16), demonstrating the 1T–OECN's ability to replicate complex neural behaviors.

Additionally, the 1T–OECN's three-terminal design enables Boolean logic operations using electrical inputs at the gate (Input X), source (Input Y), and drain (Input Z). The channel device acts like a soma, processing these signals, while the drain terminal serves as the axon (Fig. 3e). As shown in Supplementary Fig. 16, spiking occurs only when $V_G$ and $V_S$ fall within an optimal range and $I_{IN}$ exceeds a threshold. By encoding these inputs, the 1T–OECN performs six fundamental logic operations—AND, OR, XOR, NAND, NOR, and XNOR—without requiring structural modification or external circuits. Figure 3f–h illustrates the 1T–OECN's ability to execute these operations (see Supplementary Note 5 for further details).

## Scalable integration of 1T–OECN for sensory encoding and neural pathways

The compact design of 1T–OECNs enables high-density device arrays with a high geometric fill factor. To optimize space and test robustness, we fabricated vertical BBL–OECmTs on a flexible Parylene substrate (Fig. 4a), with the channel length defined by the insulator thickness (0.65 μm) and the channel width by the top source electrode opening (diameter of 15 μm, see Supplementary Fig. 18). A $1 \times 1$ cm²

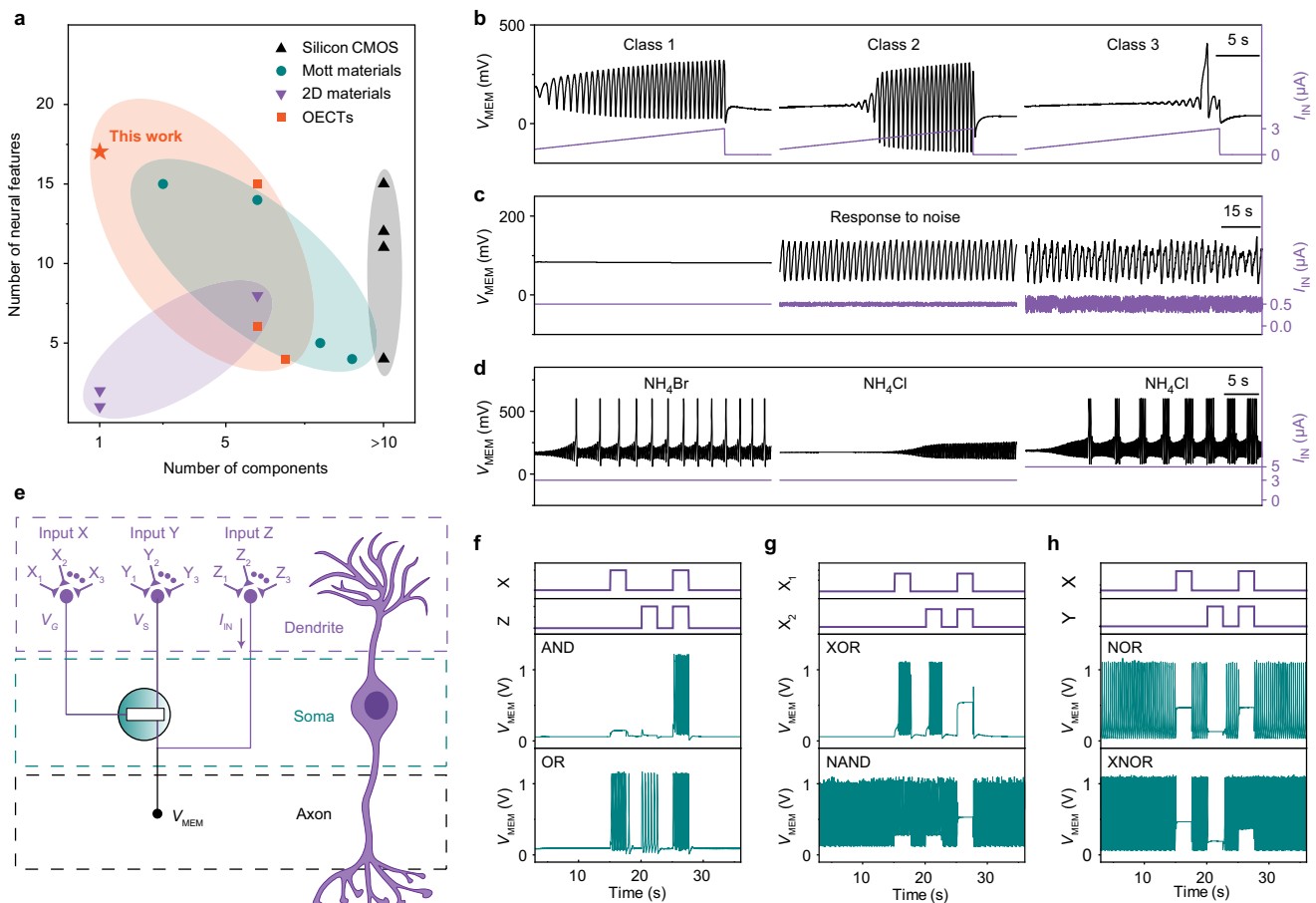

**Fig. 3 | Multifunctionality and logic operations in 1T−OECNs. a** Comparison of artificial spiking neurons across different material platforms, showing biological plausibility (number of neural features) versus circuit complexity (number of components). **b** Demonstration of the three classes of neuronal excitability: Class 1 (left), Class 2 (middle), and Class 3 (right). **c** Stochastic spiking behavior under noisy input. Left: a subthreshold input current of 0.5 μA fails to induce spiking. Middle: adding a low-amplitude noise enables consistent spiking at a defined frequency. Right: larger noise amplitude causes spike skipping while preserving the base frequency. **d** Influence of ion type and concentration on spiking behavior. Left: mixed−mode spiking with 0.1 M NH$_4$Br at $I_{IN}$ = 3 μA. Middle: tonic spiking with 0.1 M NH$_4$Cl at $I_{IN}$ = 3 μA. Right: tonic bursting with 0.1 M NH$_4$Cl at $I_{IN}$ = 5 μA. Experimental

conditions are summarized in Supplementary Table 1. **e** Morphological and functional comparison of the 1T−OECNs and biological neurons. The 1T−OECN's three terminals serve as dendrite (signal input), soma (processing), and axon (output). Inputs X, Y, and Z connect to the gate, source, and drain, respectively, while $V_{MEM}$ is measured at the drain (output terminal). **f** Implementation of AND and OR logic operations using Input X (on: 0.4 V, off: 0.26 V) and Input Z (on: 0.8 μA, off: 0.3 μA). **g** XOR and NAND logic operations using Input $X_1$ and $X_2$. For XOR: $X_1/X_2$ (on: 0.35 V, off: 0.1 V). For NAND: $X_1/X_2$ (on: 0.35 V, off: 0.2 V). **h** NOR and XNOR logic operations using Input X and Y. For NOR: X (on: 0.6 V, off: 0.3 V); Y (on: 0.05 V, off: 0 V). For XNOR: X (on: 0.6 V, off: 0.3 V); Y (on: 0.15 V, off: 0 V).

substrate contains 62,500 OECmTs (single device area of ~177 μm²), corresponding to a neuron density of 82,236 neurons/mm³ or 78,125 neurons/mg (Fig. 4b). Both these volumetric and weight-based neuron densities exceed those of the human cerebral cortex (16,000 neurons/mm³ or 12,976 neurons/mg)[32–34]. Additionally, the dimensions of each 1T−OECN are comparable to those of primary biological neurons (see Fig. 4c and Supplementary Fig. 19), further highlighting their bio−realistic design while underscoring BBL's excellent biocompatibility[35]. The coefficient of variation for the 1T−OECNs, measured across a 10 × 10 array, is 0.07 (Fig. 4d and Supplementary Fig. 20), comparable to biological neurons[36]. Under identical current stimulation and $V_G$, all 1T−OECNs in the array exhibit uniform spiking frequencies. Leveraging this uniformity, we applied 5 μA and 15 μA currents to different areas, with 5 μA serving as the background and 15 μA forming the Linköping University logo 'lı.u' (Supplementary Fig. 21a). The 1T−OECN output frequency replicates the 'lı.u' pattern, confirming stability and uniformity of the array operation (Supplementary Fig. 21b, c).

1T−OECNs can encode sensory input for neuromorphic perception, similar to how biological sensory neurons process and transmit

environmental information. Figure 4e illustrates an artificial afferent nerve consisting of a 1T−OECN and a resistive pressure sensor (see Supplementary Fig. 22 for a photograph of the artificial afferent nerve chip). When pressure is applied, the 1T−OECN spikes at a frequency proportional to the applied force (Fig. 4f and Supplementary Fig. 23). The system requires only two components, whereas traditional technologies typically use 10-20[37–39], significantly simplifying artificial nerve integration.

Beyond neuronal function, the hysteretic switching behavior of BBL−OECmTs allows them to mimic excitatory/inhibitory synaptic activity, functioning as organic electrochemical synapses (OECSs) with Hebbian learning capabilities. The connection of the OECS is shown in Fig. 4g. Figure 4h shows the long−term potentiation and depression (LTP and LTD) effects of BBL−OECSs, where the drain (postsynaptic) current increases or decreases upon stimulation with repeated 60-ms positive or negative bias pulses. Similar to biological synapses exhibiting spike-timing-dependent plasticity (STDP), the synaptic weights of BBL-OECSs can be modulated by applying time-dependent positive or negative pulses to the drain terminal (Fig. 4i). Consequently, a single device can sense stimuli, integrate information, and generate spikes

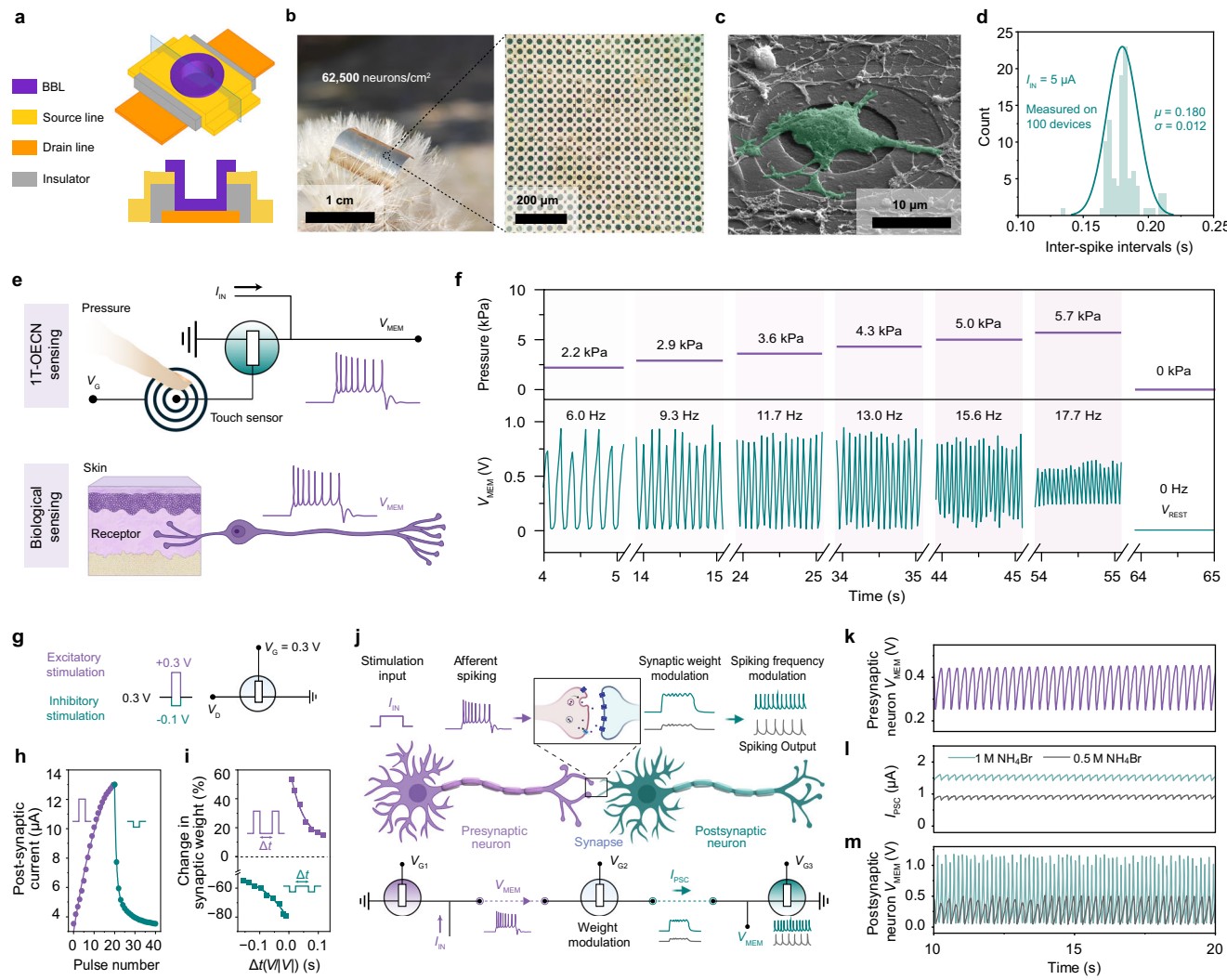

**Fig. 4 | Integrated high-density 1T–OECN arrays and applications. a** Schematic of a single 1T–OECN unit within a high-density array. **b** Photograph of a flexible 1 cm² 1T–OECN array (250 × 250, 62,500 neurons) placed on a dandelion (left), and optical image of a portion of the array (right). **c** Scanning electron micrograph (10,000×) of rat primary cortical neurons cultured on the array after 4 days in vitro ($N = 1$). **d** Spiking frequency distribution across a 10 × 10 1T–OECN array under $I_{IN} = 5\ \mu A$ and $V_G = 0.35\ V$. **e** Schematic of a pressure sensor integrated with a 1T–OECN (top), mimicking afferent nerve signaling (bottom). **f** Spiking $V_{MEM}$ output (bottom) in response to varying pressure levels applied to the touch sensor (top). **g** Schematic of the OECS. Excitatory (+0.3 V) and inhibitory (−0.1 V) pulses (60 ms duration) are applied to the drain; $V_G = 0.3\ V$. **h** Post-synaptic current as a function of pulse number. Positive pulses induce long-term potentiation (LTP), while negative pulses induce long-term depression (LTD). **i** Spike-timing dependent plasticity (STDP) characteristics of a BBL–OECS. Synaptic weight plotted as a function of time interval $\Delta t$ between positive (purple) and negative (cyan) pulses, with exponential fit (solid line). Pulse conditions as in **g**. **j** Comparison of action potential generation, synaptic modulation, and signal transmission in biological versus BBL–OECmT-based neurosynaptic networks. Spiking $V_{MEM}$ from the presynaptic neuron is modulated at the synapse and transmitted as postsynaptic current ($I_{PSC}$), which is then converted into spiking output in the postsynaptic neuron. **k** Continuous spiking $V_{MEM}$ output from the presynaptic neuron. **l** Synaptic weight modulation with 0.5 M (gray) vs 1 M (green) $NH_4Br$. **m** Post-synaptic spiking output increases in frequency and amplitude with higher synaptic weight.

like a neuron while also performing time integration and plasticity like a synapse.

We then constructed a modular neuromorphic system that integrates organic spiking neurons and synapses to replicate neural pathways, using only three OECTs (Fig. 4j). In this system, the presynaptic 1T–OECN receives external stimuli and converts them into spikes (Fig. 4k). The BBL–OECS connects two 1T–OECNs, modulating synaptic weights through ion regulation and transmitting stimuli to the postsynaptic 1T–OECN, which processes the input and generates spiking outputs at varying frequencies. Reducing the $NH_4^+$ concentration in the synapse from 1 M to 0.5 M decreases the synaptic weight by 39.5%, leading to a 39.4% reduction in postsynaptic spiking frequency under the same presynaptic input (Fig. 4l, m). This system emulates neural signal transmission, triggering spike-encoded communication while leveraging Hebbian learning for localized processing

of physiological environmental information. This approach significantly reduces complexity compared to traditional technologies, which typically require at least 10 transistors[10], and paves the way for organic neuromorphic computing circuits.

## Discussion

In conclusion, we present the development of 1T–OECNs, a novel system leveraging BBL–OECmTs. These devices exhibit wide tunability of individual states and stable on-off ratios, achieved through the integration of electrochemical memristive switching and transistor gating, enabling ion-modulated nonlinear electrical characteristics. By carefully configuring the input terminals of the OECmT, we demonstrated the realization of an ionic biomimetic artificial neuron using a single transistor, an approach not feasible with existing technologies. The 1T–OECN emulates key features of biological neurons, taking

inspiration from mechanisms such as voltage–gated activation and inactivation or delayed activation of ion channels. This device design enables the replication of high-order neural dynamics, including 17 spiking behaviors characteristic of biological neurons. The system also supports dual–mode responsiveness to both electrical and chemical signals, mirroring biological regulation mechanisms. For example, variations in the electrolyte composition modulate spiking modes, while electrical signals adjust spiking frequency. We further demonstrate how integration with pressure sensors allows the 1T–OECN to respond to sensory signals.

In addition to emulating neural dynamics, the 1T–OECN demonstrates robust computational capabilities. These include the ability to execute universal Boolean logic operations and process linearly inseparable datasets within a single transistor. The compact 1T–OECN design, with a device footprint of just 177 $\mu m^2$, enables high–density integration, achieving neuron densities exceeding 62,500 neurons/$cm^2$ while maintaining uniform spiking frequencies. This combination of ionic biomimetic functionality, computational versatility, and scalable integration highlights the potential of 1T–OECNs as a bridge between neuromorphic and traditional computing paradigms. To fully realize this potential across a broader range of biological and sensing environments, future work will focus on improving ion compatibility—for instance, by employing an extended gate[40,41] to spatially separate the sensing electrolyte from the neuron compartments-thereby enhancing both the biocompatibility and versatility of the 1T–OECN technology. These devices offer a promising platform for bioelectronics applications, including brain-machine interfaces and bio-integrative computing systems.

## Methods
### Materials
BBL was synthesized following previous reports[42]. 3-(Trimethoxysilyl) propyl methacrylate (silane A-174), ammonium bromide ($NH_4Br$), ammonium chloride ($NH_4Cl$), sodium bromide (NaBr), sodium chloride (NaCl), and methanesulfonic acid (MSA) were purchased from Sigma-Aldrich.

### Fabrication of OECmT
Planar OECTs were fabricated according to a previous protocol[42]. A schematic of the fabrication process is shown in Supplementary Fig. 24. Four-inch glass wafers were first sonicated with an industrial surfactant (2% Micro-90), acetone, and isopropyl alcohol, followed by drying with nitrogen. Electrodes consisting of 5 nm chromium and 50 nm gold were then deposited via thermal evaporation and patterned using photolithography and wet etching techniques. Next, a 1-$\mu m$-thick layer of PaC was deposited, with silane A-174 applied to enhance adhesion, serving as an insulating layer between the metal and the liquid electrolyte. An anti-adhesive layer of 2% Micro-90 surfactant was spin-coated on the PaC layer, followed by the deposition of a 2-$\mu m$-thick sacrificial PaC layer. A 5-$\mu m$-thick positive photoresist layer was then spin-coated, patterned, and developed to expose the channel area and contact pads. This photoresist layer protected the PaC layers during the subsequent plasma reactive ion etching (RIE) process (150 W, 500 sccm $O_2$, 100 sccm $CF_4$, 380 s), which removed the organic materials, including the photoresist and PaC, thus exposing the OECT channel area and contact pads while leaving the rest of the surface covered by PaC. After cleaning the entire wafer with acetone, a 20-nm-thick BBL layer was deposited by spin-coating a BBL solution (MSA, 2.5 mg/ml) onto the PaC layer, followed by water immersion and nitrogen drying. The BBL layer was then patterned through a sacrificial PaC lift-off process. Ag/AgCl paste was drop-cast on the substrate to form a 1-$\mu m$-thick, 9 $mm^2$ square gate electrode. Unless otherwise specified, 1 M $NH_4Br$ is used as the electrolyte.

Vertical OECTs were also fabricated on 4-inch wafers, following the same wafer cleaning process as planar OECTs. A schematic of the fabrication process is shown in Supplementary Fig. 25. After wafer cleaning, the first metal layer (Cr 5 nm and Au 50 nm) was evaporated onto the cleaned wafer and patterned using photolithography and wet etching. Next, a 1-$\mu m$-thick PaC layer was deposited, with silane A-174 applied to enhance adhesion, serving as an insulating layer between the first and second metal layers, and defining the length of the semiconductor channel in the vertical direction. Then, the second metal layer (Cr 5 nm and Au 50 nm) was deposited on the PaC, followed by photolithography with positive photoresist, developing, and wet etching for patterning. Another 1-$\mu m$-thick PaC layer was then deposited, with silane A-174 applied to improve adhesion, acting as an insulating layer between the electrodes and the liquid electrolyte. An anti-adhesive layer of 2% Micro-90 surfactant was spin-coated onto the PaC layer. The sacrificial PaC layer was then deposited, and the channel definition process followed the same procedure as for planar OECT fabrication.

### Fabrication of high–density 1T–OECN arrays
The microscope images of high–density 1T–OECN arrays, shown in Fig. 4b and Supplementary Fig. 19a, reveal active-matrix lines with a width of 30 $\mu m$ and a line spacing of 10 $\mu m$. The vertical and horizontal lines are defined as the source and drain lines, respectively. These lines are positioned in two parallel planes at different heights within the channel area, with a separation of ~0.65 $\mu m$, where the source line is located above the drain line. The channel is formed at the intersection of each source and drain line in the matrix, conducting from the source line downward to the drain line. The top-down view of the channel is circular, with a diameter of ~20 $\mu m$. The insulating layer between the source and drain lines defines the channel length (approximately 0.65 $\mu m$). The 3D schematic and cross-sectional view of the channel morphology are shown in Fig. 4a.

The fabrication of high–density 1T–OECN arrays on a flexible Parylene C substrate follows the process outlined below. Two-inch wafers were cleaned by sequential sonication in acetone, deionized water, and isopropyl alcohol, followed by drying with nitrogen. An anti-adhesive layer of 2% industrial surfactant Micro-90 was spin-coated, followed by the deposition of a 10-$\mu m$-thick Parylene C (PaC) layer onto the two-inch wafers. The first metal layer was processed using a lift-off technique. A 1.3-$\mu m$-thick positive photoresist was spin-coated, patterned, and developed. Then, 10 nm of Cr and 50 nm of Au were evaporated, followed by immersion in acetone and sonication. After the metal, except for the electrodes and wires, was removed, the wafer was cleaned and dried with nitrogen. Next, a 0.65-$\mu m$-thick PaC layer was deposited to serve as the insulation between the Source and Drain, defining the length of the vertical channel. Photoresist was spin-coated on the PaC layer, patterned, and developed to remove the photoresist at the endpoints of each Source line, leaving the rest of the photoresist intact. RIE etching (150 W, 500 sccm $O_2$, 100 sccm $CF_4$, 17 s) was then performed, allowing the subsequently evaporated source lines (second metal layer) to connect to the contact pads of the first metal layer for electrical testing. A 5 nm Cr and 50 nm Au layer was then evaporated, and the photoresist was spin-coated, patterned, developed, and wet etched to complete the patterning of the second metal layer (source lines). Following this, a 1-$\mu m$-thick PaC layer was deposited to provide insulation between the metal and the liquid electrolyte, preventing interference from parasitic capacitance. Positive photoresist was spin-coated, exposed, and developed, then the photoresist over the Gate and contact pad areas was etched away. An RIE etching (150 W, 500 sccm $O_2$, 100 sccm $CF_4$, 34 s) was performed to remove the PaC above the Gate and contact pad areas, facilitating subsequent electrical testing. After that, an anti-adhesive layer of 2% Micro-90 surfactant was spin-coated onto the PaC layer. And the sacrificial PaC layer was then deposited, and the channel definition process followed the same procedure as for planar OECT fabrication. After completing the fabrication process, the 10-$\mu m$-thick PaC layer can be peeled off from the 2-inch glass wafer for electrical testing.

## Electrical characterization

Electrical characterization was performed using the Keithley 4200A–SCS system (equipped with the 4225-PMU Ultra Fast I–V Module and 4225-RPM Remote Amplifier/Switch Modules). Agilent Infiniium 54832D oscilloscope was employed to capture waveforms above 10 Hz, with an internal resistance of 1 MΩ.

## Setup for sensory encoding

The circuit comprises a 1T–OECN and a commercial touch sensor (Force Sensing Resistor, FSR 402 Short, Interlink Electronics, Inc.). As shown in the photograph of the artificial afferent nerve chip in Supplementary Fig. 22, the gate terminal of the 1T–OECN is connected to the output of the touch sensor. A constant voltage ($V_G = 0.5$ V) is applied to the input of the touch sensor. The source terminal of the 1T–OECN is grounded, while its drain terminal is connected to a constant input current ($I_{IN} = 13$ μA) during testing. The resulting voltage spiking waveform ($V_{MEM}$) is measured at the drain terminal.

## Setup for neural pathways

For the neural pathway circuit depicted in Fig. 4j, three OECTs are integrated, each fulfilling a distinct functional role. The OECT positioned on the left operates as the presynaptic neuron, the central OECT functions as the synapse, and the OECT on the right serves as the postsynaptic neuron. Signal transmission is initiated by the presynaptic neuron, where the $V_{MEM}$ spiking signal is generated and subsequently transmitted to the synapse for modulation. The modulated current signal is then relayed to the postsynaptic neuron, which converts it into $V_{MEM}$ spiking signals with varying frequencies. The OECT representing the presynaptic neuron is designed with a vertical architecture, featuring channel dimensions of W = 300 μm and L = 1 μm, and employs 1 M NH₄Br as the electrolyte. The gate terminal is maintained at a constant voltage of $V_G = 0.4$ V, while the drain terminal receives a fixed input current of $I_{IN} = 10$ μA. The resultant $V_{MEM}$ signal is recorded at the drain terminal and subsequently transmitted to the Keithley 4200A–SCS system for further processing. This signal is then applied to the drain terminal of the OECT, which functions as the synapse. The synapse OECT adopts a planar architecture with channel dimensions of W = 400 μm and L = 6 μm. To facilitate synaptic weight modulation, two distinct electrolyte concentrations (0.5 M and 1 M NH₄Br) are utilized. The gate terminal is biased at a fixed voltage of 0.3 V. The postsynaptic current (PSC) is measured at the source terminal and transmitted to the Keithley 4200A–SCS system, from which it is subsequently applied to the drain terminal of the OECT, serving as the postsynaptic neuron. The OECT acting as the postsynaptic neuron is also designed with a planar architecture, with channel dimensions of W = 200 μm and L = 6 μm, and utilizes 1 M NH₄Br as the electrolyte. The gate terminal is held at a constant voltage of 0.3 V, while the source terminal is grounded. The final output, $V_{MEM}$, is recorded at the drain terminal.

## SPICE simulations

The SPICE model of the BBL–based OECmTs and 1T–OECNs was developed in B2 SPICE (EMAG Technologies). The model is designed to simulate the hysteretic behavior of BBL–based OECmTs and the spiking behavior of the 1T–OECNs. Detailed descriptions of the simulation setup and results are provided in the Supplementary Information.

## Primary neuronal culture

Primary cortical neurons were isolated from embryonic day 18 Wistar rat embryos (Javier, France). The use of primary tissues in this work was approved by the state animal ethics committee, the Landesumweltamt für Natur, Umwelt und Verbraucherschutz Nordrhein-Westfalen, Recklinghausen, Germany, under permit number 81-02.04.2023.A172. The experiments were conducted in accordance with local animal protection regulations. Cortices were

digested in cold 0.05% Trypsin EDTA (Thermo Fisher, cat. no. 25300-062) for 10 min at 37 °C followed by mechanical trituration[43] and plated at a density of 53,000 cells per cm². Prior to plating, substrates were sterilized by incubation in 70% abs. ethanol (TH.GEYER, cat. no. 2273)/ddH₂O for 20 min, followed by three washes in sterile ddH₂O. The substrates were then coated with Poly-L-Lysine (PLL; Sigma–Aldrich, cat. No. P4707-50ML) for 1 h at 37 °C, washed three times with sterile ddH₂O, and dried completely. Neurons were maintained in a B27 Plus Neuronal Culture System consisting of Neurobasal Plus medium and B27 Plus supplement (Thermo Fisher Scientific, cat. no. A3653401). The culturing medium was supplemented with 2% B27 Plus and 50 μg/ml Gentamycin (Sigma, cat. no. G1397) and maintained at 100% humidity, 5% CO₂, 37 °C. The medium was replaced 4 h after seeding, and half of the culturing medium was exchanged twice per week.

## Preparation of specimens for scanning electron microscopy and imaging

On day in vitro 4 (DIV 4), neurons were rinsed with DPBS (Thermo Fisher, cat. no.14190169), fixed in 4% v/v paraformaldehyde (Thermo Fisher, cat. no. 50-00-0) diluted in cytoskeleton-stabilizing buffer (PEM; 80 mM PIPES, 5 mM EGTA, 2 mM MgCl₂, pH 6.8) for 10 min at RT and washed three times in DPBS (5 min per wash). Samples were rinsed with 0.1 M Sodium Cacodylate buffer (NaC; Electron Microscopy Sciences, cat. no. 11652) and stabilized by incubation in 2.5% v/v Glutaraldehyde (GA; Sigma–Aldrich, cat. no. 354400)/0.1 M NaC for 2 h at RT. Samples were washed three times in 0.1 M NaC (10 min per wash). Afterward, samples were dehydrated in a gradient series of cold abs. ethanol (30% v/v, 50% v/v, 2 × 70% v/v, 3 × 95% v/v, and 2 × 100% v/v in Milli-Q H₂O). Each step was carried out for 10 min at +4 °C. The following day, samples were processed in a critical point dryer (CPD, Baltec CPD 030), in which ethanol was gradually replaced with liquid CO₂ which was let evaporate by temperature increase. Dried samples were mounted on aluminum stubs and sputtered with a layer of iridium (15 mA, 60 s) prior to imaging.

The morphology of primary cortical neurons on the chips was investigated by scanning electron microscopy (SEM). SEM imaging was performed using a Thermo Fisher (previously FEI) Magellan 400 SEM. CPD-processed samples were imaged at an acceleration voltage of 20 kV, with a working distance of 3–4.2 mm.

## Reporting summary

Further information on research design is available in the Nature Portfolio Reporting Summary linked to this article.

## Data availability

Data supporting the findings of this study are available in the paper and the Supplementary Information files. The data generated in this study are provided in the Source Data files. Source data are provided with this paper.

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

## Acknowledgements

This work was financially supported by the Knut and Alice Wallenberg Foundation (2021.0058 and Wallenberg Initiative Materials Science for Sustainability WISE), the Swedish Research Council (2020-03243, 2022-04053, 2022-04553), the European Research Council through the ERC Consolidator Grant project INFER (101125879), the European Commission through the FET-OPEN project MITICS (964677), the Pathfinder OPEN project ICONIC (101129638), and the MSCA-IF-2020 project S-OECN (101152690), the Swedish Foundation for Strategic Research (IS24-0162), and the Swedish Government Strategic Research Area in Materials Science on Functional Materials at Linköping University (Faculty Grant SFO-Mat-LiU 2009-00971). F.S. and N.S. thank Elke Brauweiler-Reuters, Elmar Neumann, and the Helmholtz Nanofabrication facility for technical support.

## Author contributions

J.J. and S.F. conceived the idea and designed the project. J.J. fabricated and characterized the 1T–OECNs. D.G., M.X., H.-Y.W., and C.-Y.Y. assisted with the fabrication of the 1T–OECNs. H.-Y.W. and C.-Y.Y. synthesized BBL. N.S., C.L.B., and F.S. cultured the primary neurons and performed SEM. D.T. performed the SPICE model simulations. J.J., D.G., D.T., and S.F. wrote the manuscript. All authors contributed to discussions and manuscript preparation.

## Funding

## Competing interests

The authors declare no competing interests.
