## [Transparent Peer Review file · Nature Communications]

Single-transistor organic electrochemical neurons

Corresponding Author: Professor Simone Fabiano

Version 0:

Reviewer comments:

Reviewer #1

(Remarks to the Author)

The authors have conducted a highly advanced study, and this paper is already complete. I recommend its publication immediately after the following minor revisions:

1. The current title appears to be designed for a broader audience. Please make it more specific to align with Nature Communications.
2. The expression "62,500 neurons/cm²" used to describe the integration density tends to be somewhat misleading by emphasizing the big number. Please include the actual dimensions of the device side by side for clarity. For example, 'the device size is 4 x 4 μm², giving a density of 62,500 neurons/cm²'
3. Adding a schematic of the device fabrication process in the Supplementary Information (SI) would be beneficial.

Reviewer #2

(Remarks to the Author)

In this work, the authors demonstrate a 1 transistor organic artificial neuron. The key device is an organic electrochemical device which displays hysteretic-like behaviour and an asymmetry between polarization and depolarization processes. With this device, a wide range of neuron-like behaviours are shown. Due to the simplified structure of the organic neuron, the overlap footprint is smaller than the other organic neurons, enabling a more aggressive scaling at the micron-scale. Nevertheless, the overall dimensions are larger than the state of the art (solid state technologies). The work is solid and important for the community as it simplifies the realization of organic neurons. I therefore recommend its publication. I only have a few comments in order to clarify some points:

1. A necessary condition for the neuron function is the hysteretic behaviour of the OECT, which requires the use of bulkier ions. If so, there is a limitation in what kind of ions can be used during operation. This means that the device cannot always act as an in situ sensor. Please comment on the main text.
2. Is it possible to show some examples of ions inducing proper spiking behaviour and some others that this is not possible?
3. Figure 3b. How different excitability classes are experimentally produced? What practically changes (in biasing or measurement configuration) from one class to another?
4. Is it necessary to have a gaussian I-V for this particular implementation? Please comment on the main text.
5. Since the footprint of the neuronal implementation is used as an argument, it would be good (if possible) to add this comparison in the form of a table for different neurons.

Version 1:

Reviewer comments:

Reviewer #2

(Remarks to the Author)

The authors addressed all my comments. This will be an important contribution to the community.

Response to the Reviewers

Dear referees, we would like to thank you for your careful reading, helpful comments, and constructive suggestions, which have significantly improved our manuscript. We have carefully considered all your comments and revised the manuscript accordingly. Below, please find our point-by-point response in red lettering to your concerns and a description of how and where revisions to the manuscript have been made.

Reviewer #1:

The authors have conducted a highly advanced study, and this paper is already complete. I recommend its publication immediately after the following minor revisions:

We thank the reviewer for their very positive commentary on our manuscript. In the following, we address their remarks:

1. The current title appears to be designed for a broader audience. Please make it more specific to align with Nature Communications.

We thank the reviewer for their suggestion. As stated in the introduction, several studies have explored the development of organic electrochemical neurons; however, all previously reported implementations rely on multiple components, including several transistors, capacitors, and resistors. In stark contrast, our work introduces a neuron design based on a single organic electrochemical transistor. With the title “*Single-transistor organic electrochemical neurons*”, our intention was to emphasize the simplicity and uniqueness of this approach. Therefore, we believe the original title remains accurate and effectively captures the core contribution of our manuscript.

2. The expression "62,500 neurons/cm²" used to describe the integration density tends to be somewhat misleading by emphasizing the big number. Please include the actual dimensions of the device side by side for clarity. For example, 'the device size is 4 x 4 μm^2 , giving a density of 62,500 neurons/cm²'

Excellent comment. We have included the actual device dimensions alongside the integration density in the abstract and introduction of the revised manuscript to provide clearer context.

3. Adding a schematic of the device fabrication process in the Supplementary Information (SI) would be beneficial.

Again, excellent comment. We have included a schematic of the device fabrication process in Supplementary Figs. 24 and 25 of the revised manuscript.

Reviewer #2:

In this work, the authors demonstrate a 1 transistor organic artificial neuron. The key device is an organic electrochemical device which displays hysteretic-like behaviour and an asymmetry between polarization and depolarization processes. With this device, a wide range of neuron-like behaviours are show. Due to the simplified structure of the organic neuron, the overlap

footprint is smaller than the other organic neurons, enabling a more aggressive scaling at the micron-scale. Nevertheless, the overall dimensions are larger than the state of the art (solid state technologies). The work is solid and important for the community as it simplifies the realization of organic neurons. I therefore recommend its publication. I only have a few comments in order to clarify some points:

We thank the reviewer for their very positive commentary on our manuscript. In the following, we address their remarks:

1. A necessary condition for the neuron function is the hysteretic behaviour of the OEET, which requires the use of bulkier ions. If so, there is a limitation in what kind of ions can be used during operation. This means that the device cannot always act as an in situ sensor. Please comment on the main text.

We thank the reviewer for raising this important point. We agree that a necessary condition for the neuron's function is the hysteretic behavior of the OEET, which is promoted by specific interactions between the electrolyte ions and the BBL backbone. In our case, hysteresis is observed in the presence of electrolytes capable of forming hydrogen bonds, such as those containing amines, carboxylic acids, or protons in mildly acidic conditions. These interactions appear to facilitate ionic trapping or accumulation, which underpins the observed spiking behavior. As a result, not all electrolytes support spiking, and we acknowledge that this imposes a limitation on the kinds of ions that can be used during operation. However, our experiments show that several electrolytes commonly found in biological systems, such as ascorbic acid and glutamic acid at physiologically relevant concentrations, can reliably induce spiking in the 1T-OECN. For example, as shown in Fig. R1, the 1T-OECN responds in real time to varying concentrations of ascorbic acid within the biologically relevant range of 150-300 μM , with spiking frequency increasing with concentration.

Fig. R1. Real-time sensing response of 1T-OECNs to varying concentrations of ascorbic acid within a biologically relevant range

While these results demonstrate that neuron-like behavior can be achieved in realistic ionic environments, we acknowledge that compatibility with all ionic species cannot be guaranteed. To address this, we are currently exploring a decoupled sensing strategy using an extended gate architecture. This approach would isolate the sensing and neuron compartments, allowing independent tuning of the ionic environments for optimal sensing and spiking performance. This strategy will be a focus of our future work. In response, we have added a discussion of this

limitation and our proposed solution at pages 13-14 of the revised manuscript.

2. Is it possible to show some examples of ions inducing of proper spiking behaviour and some others that this is not possible?

Excellent comment. We tested a variety of electrolytes and found that the 1T-OECN exhibits diverse spiking behaviors depending on the ionic composition. Electrolytes capable of forming hydrogen bonds, such as those containing amines, carboxylic acids, or protons in mildly acidic environments, consistently induced spiking. In contrast, simple monovalent and divalent salts such as NaCl, KCl, CsCl, MgCl₂, and CaCl₂ did not. In addition to the ionic composition, device geometry also influences spiking. Under identical ionic conditions (e.g., 1 M NH₄Cl), spiking occurred in devices with a 100 μm × 10 μm channel but not in those with a 1000 μm × 100 μm channel, despite having the same W/L ratio and similar transfer characteristics. This indicates that intrinsic device parameters, such as capacitance and charging dynamics, play a critical role in enabling spiking. Together, these results highlight that spiking in 1T-OECNs is governed by a combination of ionic and geometric factors, and a strict binary classification of electrolytes is not immediately applicable.

3. Figure 3b. How different excitability classes are experimentally produced? What practically changes (in biasing or measurement configuration) from one class to another?

Excellent comment. The different excitability classes shown in Fig. 3b were experimentally achieved by adjusting the biasing conditions of the 1T-OECN, specifically the values of V_G and V_S . These parameters were tuned to generate Class 1, Class 2, and Class 3 spiking modes, as defined in the manuscript. The specific configurations used to produce each spiking mode are summarized in the original Supplementary Table 1, which outlines the experimental conditions for the various spiking behaviors presented in Fig. 3b. We acknowledge that, due to the absence of a direct reference in the main text, this table may have been easily overlooked. In response to the reviewer's suggestion, we have added a note in the caption of Fig. 3 directing readers to Supplementary Table 1 for complete details on the corresponding experimental conditions.

4. Is it necessary to have a gaussian I-V for this particular implementation? Please comment on the main text.

As already discussed in the main text, the operation of the 1T-OECN relies on the asymmetric transient response that arises from the ion-tunable antiambipolarity of the BBL-based OECT. The resulting Gaussian-shaped $I-V$ transfer characteristic plays a critical role in defining the voltage window over which the device transitions between low- and high-resistance states in a hysteretic manner. In this particular implementation, the Gaussian $I-V$ profile is not merely a feature but a functional requirement. It enables the device to perform thresholding and spiking operations, as the non-monotonic $I-V$ behavior provides the necessary switching dynamics analogous to biological neuronal excitability. The spiking mechanism is inherently linked to the transitions across the peak of the Gaussian curve, where the balance between fast and slow transient responses enables action potential generation. To clarify this further, we have added a short note on p.4 of the revised manuscript reinforcing the need for the Gaussian $I-V$ behavior in our current implementation.

5. Since the footprint of the neuronal implementation is used as a argument, it would be good (if possible) to add this comparison in the form of a table for different neurons.

We thank the reviewer for this excellent suggestion. In the revised manuscript, we have added a column in Supplementary Table 2 that provides a detailed comparison of footprint sizes reported across various studies involving different artificial neuron technologies. While some prior works, particularly those based on more mature platforms such as silicon transistors or

inorganic memristors, have demonstrated smaller device footprints, the 1T-OECN presented here sets a record for ion-based artificial neuron technologies. It is important to note that the fabrication process strongly influences the device footprint. The use of advanced lithography systems, such as deep-UV or electron-beam lithography, could enable further miniaturization of the 1T-OECN. We are currently exploring such process optimizations to fully leverage the inherent design simplicity of the 1T-OECN, with the aim of further reducing the footprint in future studies.

Response to the Reviewers

Dear referees, we would like to thank you for your careful reading, helpful comments, and constructive suggestions, which have significantly improved our manuscript. We have carefully considered all your comments and revised the manuscript accordingly. Below, please find our point-by-point response in red lettering to your concerns.

Reviewer #2:

The authors addressed all my comments. This will be a important contribution to the community.

We thank the reviewer for their very positive commentary on our manuscript.